# Molecular Cardiac Changes in Feline Hyperthyroidism and Hypertrophic Cardiomyopathy: Focus on Desmin, Calreticulin, and Interleukin-10 Expression

**DOI:** 10.3390/ani15121719

**Published:** 2025-06-10

**Authors:** Izabela Janus-Ziółkowska, Joanna Bubak, Massimiliano Tursi, Cristina Vercelli, Rafał Ciaputa, Małgorzata Kandefer-Gola, Agnieszka Noszczyk-Nowak

**Affiliations:** 1Department of Pathology, Wrocław University of Environmental and Life Sciences, 50-375 Wrocław, Poland; joanna.bubak@upwr.edu.pl (J.B.); rafal.ciaputa@upwr.edu.pl (R.C.); malgorzata.kandefer-gola@upwr.edu.pl (M.K.-G.); 2Department of Veterinary Science, University of Turin, 10095 Grugliasco, Italy; massimiliano.tursi@unito.it (M.T.); cristina.vercelli@unito.it (C.V.); 3Department of Internal Diseases with Clinic of Dogs, Cats and Horses, Wrocław University of Environmental and Life Sciences, 50-366 Wrocław, Poland

**Keywords:** hyperthyroidism, hypertrophic cardiomyopathy, feline, immunohistochemistry, heart

## Abstract

Hyperthyroidism is an important endocrine disorder in adult and old cats, affecting multiple organs, with the heart developing a hypertrophic cardiomyopathy phenotype. Cardiac hypertrophy may resolve with treatment, but structural alterations persist. This study was conducted to assess the changes in proteins present in healthy and affected myocardium and based on left ventricular specimens obtained from cats with hyperthyroidism, hypertrophic cardiomyopathy, and from healthy cats. The immunohistochemical analysis presented both hyperthyroidism and hypertrophic cardiomyopathy groups with a significant loss of contractile protein (desmin) and an increase in interleukin-10, a cytokine characterised by anti-inflammatory properties and activated in myocardial remodelling. These changes were mainly related to cardiomyocyte degeneration and narrowing of the coronary arteries, and not to the primary disease.

## 1. Introduction

Feline hyperthyroidism (FHT) is the most frequent endocrinopathy in middle-aged or older cats without sex predisposition [1,2,3,4,5]. The clinical presentation of the disease is various due to a broad role of thyroid hormones in the organism [3,5]. One of the organs commonly affected during FHT is the heart. The elevation of thyroid hormones and elevation in cardiac metabolic rate may lead to the development of a hypertrophic cardiomyopathy (HCM) phenotype with thickening of the left ventricular wall and interventricular septum. This may lead to the subsequent enlargement of the left atrium and left-sided heart failure [6,7]. The effect of thyroid hormones on cardiac tissue is complex and can be associated with a high-output state resulting from increased tissue metabolism, changes in under- or overexpression of various genes responsible for cardiomyocyte structure and function, or positive inotropic, dromotropic, and chronotropic effects of the upregulation of cardiac beta-1 receptors [3,8,9].

Although the left ventricular wall and interventricular septum may show normal thickness in cats with treated FHT, the degree of histopathological changes is similar to cats with hypertrophic cardiomyopathy that show a thickened left ventricular wall and interventricular septum [10]. The most important structural changes involve cardiomyocyte degeneration with fragmentation of the sarcoplasm and a decrease in the coronary artery lumen presenting as lower lumen-to-area ratio (LAR), therefore with an increase in the thickness of the arterial wall [10]. The advanced histopathological changes in cats with FHT presenting normal ventricular dimensions point to the need for further studies on structural disturbances in hyperthyroidism, including cats with treated disease. To further classify changes that occur in the myocardium of cats with FHT, we conducted research on the immunoreaction of desmin, calreticulin, and interleukin-10 (IL-10) in cats with FHT and HCM.

Desmin is the main component of the cardiomyocyte cytoskeleton, forming intermediate filaments and building intercalated discs. It plays an important role in the normal contractile function of the myocardium. Heart failure is accompanied by changes in the cardiomyocyte structure, the loss of cross-striation, and the disorganisation of desmin filaments [11,12,13,14].

Calreticulin is a calcium (Ca^2+^)-binding protein essential for cardiac development, but its expression is downregulated following birth [15,16,17,18]. Although the overexpression of calreticulin in cardiomyocytes induces dilated cardiomyopathy (DCM) in animals, elevated levels of anti-calreticulin antibodies were noted in the sera of patients with both DCM and HCM [19]. It was also hypothesised that calreticulin may have a protective effect on the heart in cases of cardiac hypertrophy [20]. On the other hand, research suggests a role of calreticulin overexpression in cardiac fibrosis [21].

Interleukin-10 is a member of the cytokine family responsible for inhibiting T cells, monocytes, and macrophages, thus playing an anti-inflammatory role [22,23,24,25]. It improves cardiac remodelling after myocardial infarction, protects the ischemic heart from reperfusion injury, attenuates myocarditis, improves heart function, and reduces hypertrophic remodelling, but also promotes the stiffness of the heart by enhancing its fibrosis [22,25,26,27,28,29,30,31,32].

Only desmin expression has been evaluated so far in a small group of HCM cats [33]. None of the abovementioned proteins was examined in FHT-related myocardial changes. The aim of the current research was to compare the immunoreaction of desmin, calreticulin, and IL-10 in left ventricular specimens obtained from cats with HCM and FHT and to relate the protein expression changes to alterations in cardiac gross and histopathological examination.

## 2. Materials and Methods

### 2.1. Animals

Hearts obtained during necropsy from 38 owned cats served the purpose of the study. The necropsies were performed in the Department of Pathology (Wrocław University of Environmental and Life Sciences; Poland) or in the Department of Veterinary Sciences (University of Turin; Italy). Animals’ euthanasia was performed with intravenous injection of pentobarbital solution. To allow the procedure and sample collection, all animal owners signed an informed consent prior to the necropsy. According to the national law, studies conducted on tissue samples obtained from necropsy do not require approval from the ethics committee.

Animals were divided into three groups, based on clinical information: hyperthyroidism (FHT group; *n* = 16), hypertrophic cardiomyopathy (HCM group; *n* = 12) and control (*n* = 10) groups, as described previously [10]. Appropriate guidelines [5,6] served for the diagnosis and treatment of animals in FHT and HCM groups [10]. The details of inclusion/exclusion criteria are provided below. Animals included in the control group had a clinical exclusion of cardiac disease, hyperthyroidism, and systemic hypertension (based on echocardiographic examination, blood examination, and systemic pressure measurement, respectively). Their euthanasia was a result of severe trauma. Animals included in the FHT group showed elevation of thyroid hormone levels (based on blood examination) with or without other clinical signs of FHT. The presence or absence of cardiac hypertrophy and/or signs of heart failure were not inclusion/exclusion criteria in this group. All cats in the FHT group received pharmacological treatment from the moment of diagnosis to the moment of euthanasia. Cats in the HCM group were diagnosed on the basis of echocardiographic examination (the inclusion criteria were diastolic interventricular septum thickness and/or left ventricular posterior wall thickness exceeding 6 mm). Also, other causes of left ventricular hypertrophy (hyperthyroidism, chronic kidney disease, systemic hypertension, neoplastic disease) were ruled out on the basis of appropriate examination. When developing signs of heart failure, cats in that group received appropriate treatment. All cats in the HCM group developed stage C (symptomatic [6]) disease prior to euthanasia.

As the results of histopathological examination of the hearts in FHT and HCM are various, often overlapping, and therefore unspecific [10,34], and there are no clear cut-off histopathological criteria for FHT and HCM, they were not taken into account in the formation of study groups.

Following necropsy and fixation in 7% buffered formalin for 24 h, the hearts underwent detailed pathomorphological examination, described in the previous study [10], followed by immunohistochemical examination.

### 2.2. Pathomorphological Examination

The cardiac measurements (heart weight, heart width, left atrial appendage height and width, left ventricular wall thickness) and left ventricular histopathological examination results (cardiomyocyte degeneration, myocardial inflammatory infiltration, myocardial disarray, cardiomyocyte diameter, myocardial fibrosis, coronary arteries LAR) were retrieved from the previous study [10] and used in the statistical analysis of the current results.

### 2.3. Immunohistochemical Examination

Freshly cut 3 µm-thick sections of formalin-fixed, paraffin-embedded tissues were mounted on microscope slides (Epredia; Breda, Netherlands). Staining was performed on a Leica Bond-Max (Leica Biosystems; Newcastle upon Tyne, UK) according to the following protocol. First, tissues were deparaffinised (Bond Dewax Solution, Leica Biosystems; Newcastle upon Tyne, UK) and pre-treated with the Bond Epitope Retrieval Solution 1 or Bond Epitope Retrieval Solution 2 (Leica Biosystems; Newcastle upon Tyne, UK): calreticulin: ER1 for 30 min; IL-10: ER1 for 20 min; desmin: ER2 for 20 min. The activity of the endogenous peroxidase was blocked by Peroxide Block using the BOND Polymer Refine Detection System (Leica Biosystems; Newcastle upon Tyne, UK). In order to measure the level of the studied antigens, the antibodies were applied for 15 min at room temperature in the following concentrations: calreticulin: 1:500 (Abcam, Cambridge, UK); IL-10: 1:100 (Cloud-Clone Corp.; Katy, USA); desmin clone D33-RTU (CP Lab Safety, Novato, CA, USA). The next steps involved the dilution of the antibodies in the Bond Primary Antibody Diluent (Leica Biosystems; Newcastle upon Tyne, UK), followed by sample incubation with Post Primary and Polymer using the BOND Polymer Refine Detection System (Leica Biosystems; Newcastle upon Tyne, UK). The 3, 3′-diaminobenzidine (DAB Chromogen; Leica Biosystems; Newcastle upon Tyne, UK) served as a substrate for the reaction, and all the sections were counterstained with haematoxylin (BOND Polymer Refine Detection System, Leica Biosystems, UK). Negative controls were generated in the absence of the primary antibodies.

The immunohistochemical reaction was preliminary evaluated by four pathologists experienced in immunohistochemical studies to assess the accuracy of the reaction.

For the immunohistochemical analysis, ten randomly chosen view fields, magnified 200× (total area 2.78 mm^2^) and obtained from slides stained with each antibody, were photographed using a Leica DM500 microscope coupled with a Leica ICC50W camera (Leica Microsystems, KAWA.SKA; Zalesie Górne, Poland). Binary images were obtained using ImageJ software (ver. 1.53a; LOCI, University of Wisconsin, Madison, WI, USA), to differentiate the positive reaction (black) from non-reacting tissue (white) (Figure 1).

The calculation of the percentage of black area per total image area served as a proportion of positive tissue. A mean value was calculated for each specimen. Additionally, slides for desmin reaction were evaluated using Olympus CX41 microscope (Olympus, Tokyo, Japan) to assess the immunoreaction pattern within the disarray areas. The evaluated patterns included normal, loss of cross-striation and intercalated discs, granular appearance, and presence of longitudinal staining [35].

### 2.4. Statistical Analysis

The statistical analysis was performed using Statistica 13.3 (Tibco Inc, Poland) and appropriate tests. Data normality was tested using Shapiro–Wilk test. Due to a non-normal distribution of data, the difference between the groups was tested using Kruskal–Wallis analysis with Dunn post-hoc test. The correlation between the immunohistochemical reaction and cardiac pathomorphological analysis (gross and histopathological) was tested using Spearman correlation test. The correlation strength was assessed as poor (0.1 ≤ *r* < 0.3), fair (0.3 ≤ *r* < 0.6), moderate (0.6 ≤ *r* < 0.8), very strong (0.8 ≤ *r* < 1), or perfect (*r* = 1) [36]. The significance level was set at *p* < 0.05.

## 3. Results

The study group comprised 38 animals weighing from 3 to 8 kg (median 4.5 kg). Fifty-five percent of all animals were males. The majority of cats (63.2%) were domestic shorthair cats; among other breeds, the most numerous were Maine Coon (15.7%) and British shorthair (10.5%). Norwegian Forest, sphynx, Scottish fold, and ragdoll breeds were represented by one cat each. The animals were aged from 1 to 20.5 years (median 9 years), with higher ages in the FHT group (median 15 years, range 4–20.5 years) than in the HCM group (median 5 years; range 3–15 years) and control group (median 1.5 years; range 1–17 years).

The morphometric results obtained in each group are presented in Table 1.

Six cats in the FHT group (37.5%) presented with mild left ventricular hypertrophy (cardiac weight > 20 g) at the time of death or euthanasia.

The representative examples of the immunohistochemical reaction for studied antibodies in each group are presented in Figure 2, Figure 3 and Figure 4.

The desmin reaction was present in the cytoplasm of cardiomyocytes; IL-10 showed an intense reaction in single immune cells dispersed within the myocardium and a less intense cytoplasmic immunoreaction within the cardiomyocytes. The calreticulin reaction was visible in the interstitial tissue.

The percentage of the desmin-positive area was significantly lower in both the HCM and FHT groups as compared to control group (*p* = 0.0002 and *p* = 0.0003, respectively; Figure 5). The percentage of the calreticulin-positive area was significantly higher in the HCM group than in the control group (*p* = 0.01; Figure 5) but did not differ between the FHT group and control group (*p* = 0.18; Figure 5). The percentage of the IL-10-positive area was significantly higher in the HCM group than in the control group (*p* = 0.03; Figure 5) and in the FHT group than in the control group (*p* = 0.03; Figure 5). Nonetheless, as presented in Figure 5, an overlap existed between the results for calreticulin and IL-10 obtained in the study groups and in the control group. Specimens from the HCM and FHT groups showed no significant differences in any of the evaluated reactions (*p* = 0.75 for IL-10; *p* = 0.75 for calreticulin; *p* = 0.94 for desmin).

All specimens from the control group showed a normal pattern of desmin immunoreaction in the areas of physiological disarray of interweaving fibres (Figure 2B) with visible cross-striation and desmosomes. Among the HCM specimens, one showed a normal desmin immunoreaction, three specimens showed only the loss of cross-striation, and eight specimens showed a concomitant loss of striation and granular appearance of the desmin reaction (Figure 3B). Among the FHT specimens, five specimens showed only a loss of cross-striation (Figure 4B), and eleven specimens showed a concomitant loss of striation and granular appearance of the desmin reaction. In none of the specimens was the longitudinal arrangement of filaments noted.

The desmin-positive area showed a negative correlation with heart weight (moderate: *p* < 0.05; *r* = −0.77), heart width (fair: *p* < 0.05; *r* = −0.49), left atrial appendage height (fair: *p* < 0.05; *r* = −0.59) and width (moderate: *p* < 0.05; *r* = −0.64), and left ventricular cardiomyocyte degeneration score (moderate: *p* < 0.05; *r* = −0.62); additionally, the desmin-positive area showed a fair positive correlation with left ventricular coronary artery LAR (*p* < 0.05; *r* = 0.55). The calreticulin-positive area showed a fair positive correlation with heart weight (*p* < 0.05; *r* = 0.54). The interleukin-10-positive area showed a fair positive correlation with left ventricular cardiomyocyte degeneration (*p* < 0.05; *r* = 0.50) and a fair negative correlation with left ventricular coronary artery LAR (*p* < 0.05; *r* = −0.50). No correlations were found between myocardial fibrosis and the expression of any examined proteins (*p* > 0.05 for all analyses).

## 4. Discussion

Cats with HCM and FHT present similar changes in protein expression, despite the primary cause of myocardial changes.

Our previous study [10] demonstrated that, although cats with FHT exhibit normal left ventricular wall thickness, they present histopathological alterations in the left ventricle similar to those observed in cats with HCM. The current study further indicates that, in terms of protein expression changes, there are no significant differences between cats with HCM and those with FHT.

In human and canine DCM-related heart failure and in canine ischemia-related progressive heart failure, the structure of cardiomyocytes and intercalated discs undergo structural disorganisation, leading to the loss of myofilaments and their supporting proteins, such as desmin [14,37]. In atria of dogs with DCM, cardiomyocytes show less prominent desmin expression and less prominent desmosome visibility. However, the overall percentage of the positive staining area is similar to healthy dogs [38]. The present study suggests that the loss of contractile proteins is not specific to the underlying cardiac disease. The results suggest that it might be related to cardiac hypertrophy, cardiomegaly, left atrial enlargement, and compromised blood supply to cardiomyocytes due to arterial narrowing rather than to primary disease.

Myocardial disarray is the characteristic histopathological feature of both human and feline HCM [10,35,39]. In humans with HCM, areas of disarray present a specific pattern of desmin immunoreactivity characterised by a decrease or loss in the labelling of intercalated discs and Z-bands, longitudinal arrangement of desmin filaments, and intense granular staining in occasional myocytes [35,40]. However, this pattern was not observed previously in feline HCM [33]. Consistent with these findings, our study did not reveal the complete combination of these alterations. Instead, we observed only the loss of cross-striation and intercalated discs, along with a granular staining pattern. Notably, these changes were present in both the HCM and FHT groups.

Calreticulin plays multiple and sometimes counteracting roles in the context of cardiac hypertrophy [17,20]. It can have a protective effect on the heart: calreticulin expression increases in a mouse model of aortic stenosis-induced cardiac hypertrophy and calreticulin-null cardiomyocytes show more intense hypertrophy as compared to cells containing calreticulin [20]. Calreticulin increases cardiomyocyte endoplasmic reticulum/sarcoplasmic reticulum Ca^2+^ capacity and mechanical work potential, but at the same time may lead to cardiomyopathy, reduce cell–cell communication, and induce increased collagen deposition and cardiac fibrosis [17]. The mechanism of calreticulin-induced fibrosis is most probably related to mechanical stretch of cardiac fibroblasts caused by cardiomyocytes with enhanced efficiency [17]. In the current research, only the HCM group showed significantly higher levels of calreticulin, as compared to the control group. Although the FHT group showed an elevation in calreticulin immunoreaction, it did not reach significance. This can be related to the heterogenicity of the FHT group involving cats with a normal or hypertrophied heart. Due to the small number of animals in the FHT subgroups (with hypertrophy and without hypertrophy), a deeper statistical analysis was not possible.

Although calreticulin immunoreactivity was elevated in HCM group and was correlated with heart weight in the current research, the protein was present only in the interstitial tissue. The interstitial tissue plays a supportive role for cardiomyocytes, but heart failure is accompanied by an increased amount of interstitial tissue and subsequent myocardial fibrosis and stiffness [14,37], also in the course of HCM [34]. The role of calreticulin in cardiac fibrosis remains debated. Fibroblasts presenting increased calreticulin expression show higher viability and invasiveness and decreased apoptosis, suggesting a potential role of calreticulin in myocardial fibrosis [21]. At the same time, calreticulin was not related to cardiac fibrosis in the animal model, although it was upregulated in the animal model of lung fibrosis [41]. In our previous study [10], the FHT and HCM groups did not show more severe fibrosis, as compared to the control group, despite all cats in the HCM group presenting with heart failure. Also, no correlation between fibrosis and calreticulin expression was noted. Further studies may evaluate the presence of calreticulin in chronic heart failure with significant fibrosis to better explain the role of this protein in naturally occurring diseases.

Interleukin-10 is produced mainly by monocytes, dendritic cells, macrophages, regulatory T cells, and B cells [22]. However, under pathological conditions, IL-10 can also be produced by smooth muscle or cardiac muscle cells [42,43,44]. IL-10 shows anti-inflammatory activity but also possesses profibrotic potential as it promotes myofibroblast proliferation and collagen production [22,25,45,46]. In the current study, no relation between IL-10 expression and the extent of fibrosis was noted. IL-10 is thought to limit pressure overload-induced cardiac remodelling [29], and it may counteract some of the deleterious effects of TNF-alpha in inflammatory response [47]. However, the majority of research is conducted on acute disease models [29,31,43,44]. In the current study, evaluating the expression of IL-10 in chronic animal disease conditions, protein immunoreactivity was found mainly in cardiomyocytes with single IL-10-positive inflammatory cells dispersed within the myocardial interstitial tissue. A negative correlation of IL-10 expression and coronary arterial lumen size, together with a positive correlation of protein expression and cardiomyocyte degeneration, points to a possible protective role played by IL-10 in cardiomyocyte injury caused by chronic impaired coronary blood flow and subsequent cellular degeneration.

FHT excludes the diagnosis of primary HCM in cases of echocardiographically noted concentric hypertrophy [6]. Nonetheless, one cannot rule out the concomitant occurrence of HCM and FHT. Also in our study, 37.5% of cats with FHT presented with cardiac hypertrophy that could be caused by primary HCM. There are no clear cut-off histopathological criteria to distinguish cats with HCM and FHT [34], as even the disarray, considered the most typical sign of HCM, can be observed in other causes of ventricular hypertrophy [10,48]. Moreover, the histopathological picture of HCM is very complex and can involve not only cardiomyocyte hypertrophy but also fibrosis, inflammatory response, or other types of myocardial degeneration, making the diagnosis even more challenging [34]. Dividing the FHT group into subgroups of cats with concomitant hypertrophic phenotype and cats with ventricular measurements within the normal range could possibly add to the results and conclusions of the study. Nonetheless, the subgroups were too small to obtain a reliable statistical analysis. Collecting sufficient-size groups is challenging and requires collaboration between clinicians and pathologists to provide a detailed clinical history followed by pathological examination. Therefore, it can limit the results obtained in the study.

## 5. Conclusions

Feline hyperthyroidism induces significant alterations in myocardial protein expression in the myocardium. Similarly to changes in histopathological structure noted in the previous study, the loss of contractile proteins and increased IL-10 immunoreactivity observed in the current research are similar to changes reported in HCM. These alterations may be more closely associated with secondary processes such as cardiac hypertrophy, cardiomyocyte degeneration, and reduced myocardial perfusion, observed in both groups of animals, rather than with the primary aetiology of myocardial disease. Due to a possibility of concomitant FHT and HCM, further studies involving larger groups of animals are required to better describe changes occurring in feline hearts during cardiac hypertrophy with various aetiology.

## Figures and Tables

**Figure 1 animals-15-01719-f001:**
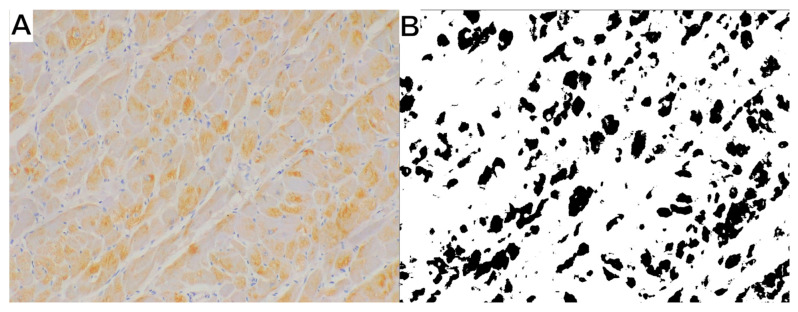
The method of evaluating the immunohistochemical reaction. (**A**) Original image presenting interleukin-10 immunoreaction; magnification 200×. (**B**) Binary image obtained from the original image: positive reaction—black; non-reacting tissue and background—white.

**Figure 2 animals-15-01719-f002:**
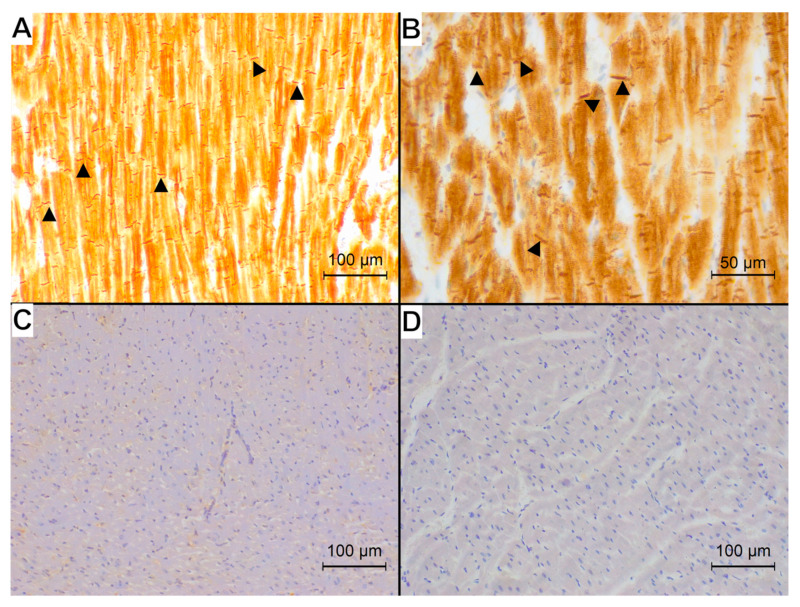
Immunohistochemical reaction (brown) of examined proteins observed in control group. (**A**) Desmin immunoreaction visible in the cardiomyocytes: intercalated discs and Z-bands (examples marked with arrowheads) show intense immunoreaction with prominent cross-striation of cardiomyocytes; magnification 200×. (**B**) Desmin immunoreaction in an area of physiological disarray showing no alterations; magnification 400×. (**C**) Faint calreticulin immunoreaction visible in the interstitial tissue; magnification 200×. (**D**) Interleukin-10 immunoreaction in control group—no staining within the cardiomyocytes; magnification 200×.

**Figure 3 animals-15-01719-f003:**
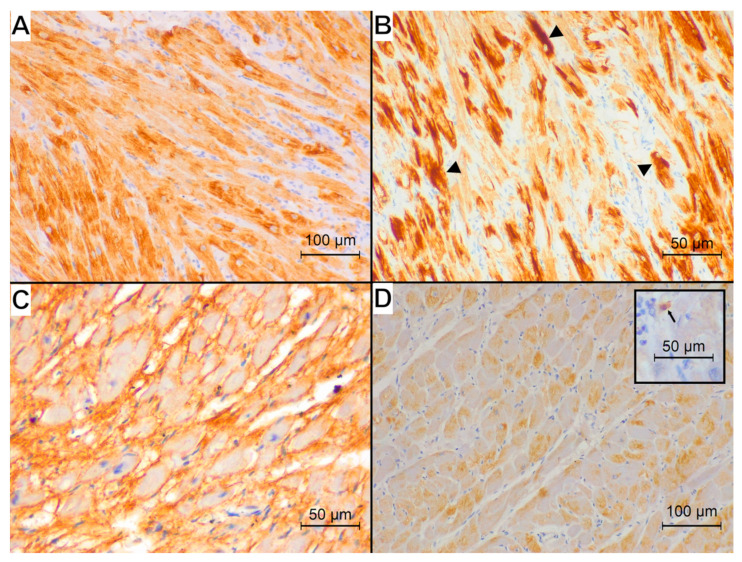
Immunohistochemical reaction (brown) of examined proteins observed in HCM group. (**A**) Desmin immunoreaction visible in the cardiomyocytes: loss of reaction in intercalated discs and Z-bands and subsequent loss of cross-striation; magnification 200×. (**B**) Desmin immunoreaction in an area of disarray showing loss of striation and granular appearance of desmin reaction (arrowheads); magnification 400×. (**C**) Calreticulin immunoreaction visible in the interstitial tissue; magnification 400×. (**D**) Interleukin-10 immunoreaction: moderate staining within the cardiomyocytes and single strongly stained immune cells dispersed within the myocardium (box; arrow; magnification 400×); magnification 200×.

**Figure 4 animals-15-01719-f004:**
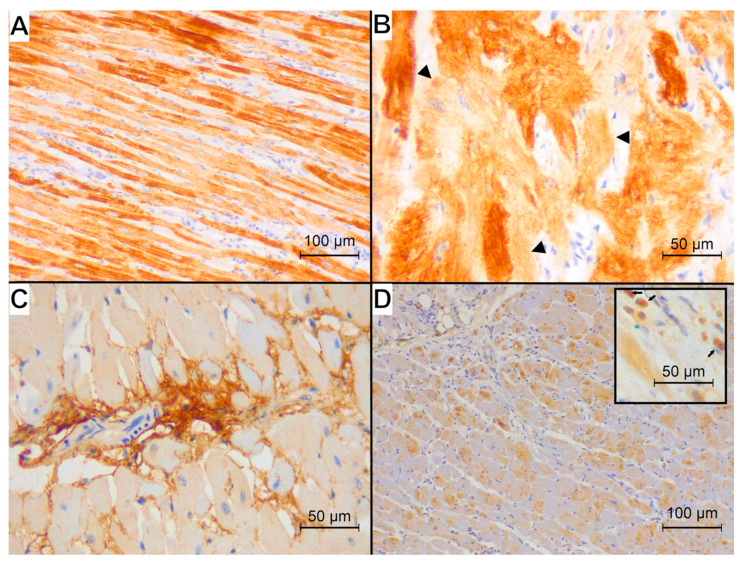
Immunohistochemical reaction (brown) of examined proteins observed in the FHT group. (**A**) Desmin immunoreaction visible in the cardiomyocytes: loss of reaction in intercalated discs and Z-bands resulting in loss of cross-striation; magnification 200×. (**B**) Desmin immunoreaction in an area of disarray showing loss of striation (arrowheads); magnification 400×. (**C**) Calreticulin immunoreaction visible in the interstitial tissue; magnification 400×. (**D**) Interleukin-10 immunoreaction: moderate staining within the cardiomyocytes and single strongly stained immune cells dispersed within the myocardium (box; arrows; magnification 400×); magnification 200×.

**Figure 5 animals-15-01719-f005:**
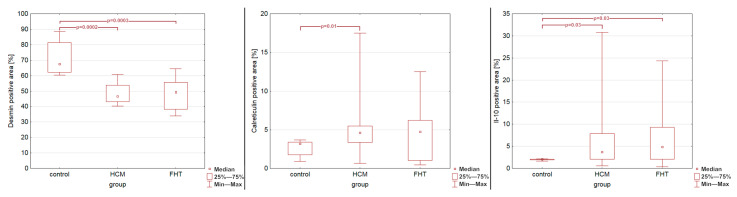
The percentage of positive area of examined proteins: desmin, calreticulin, and interleukin-10 in the control group and in cats with hypertrophic cardiomyopathy (HCM) or hyperthyroidism (FHT). Statistically significant differences are marked on the charts.

**Table 1 animals-15-01719-t001:** The morphometric analysis of the studied hearts.

Group/Parameter	Control *n* = 10	FHT *n* = 16	HCM *n* = 12
Heart weight [g] median (min–max)	15 (13–15)	22 (18–24)	39 (30–45)
Heart height [mm] median (min–max)	38.2 (30.3–41.2)	40.4 (36.6–53.6)	52.1 (40.3–63.5)
Left atrial appendage height [mm] median (min–max)	10.9 (8.2–13.2)	15.9 (10.0–29.8)	26.6 (19.5–38)
Left atrial appendage width [mm] median (min–max)	10.4 (8.4–11.8)	15.9 (12.5–16.9)	20.8 (12.8–34.2)
Interventricular septum thickness [mm] median (min–max)	5.9 (4.9–6.8)	6.0 (3.1–8.4)	8.7 (6.3–9.6)
Left ventricular posterior wall thickness [mm] median (min–max)	6.4 (6.0–7.9)	6.5 (4.0–9.9)	10.2 (7.9–13)

Data are presented as median (range).

## Data Availability

The data used in the study are available within the manuscript and at the corresponding author.

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
