# Peer review of "Molecular Cardiac Changes in Feline Hyperthyroidism and Hypertrophic Cardiomyopathy: Focus on Desmin, Calreticulin, and Interleukin-10 Expression"

_animals, 2025, doi:10.3390/ani15121719_

Round 1
Reviewer 1 Report
Comments and Suggestions for Authors
Dear authors,
Your manuscript contains important and unique information about cardiac pathology in feline hyperthyroidism/hypertrophic cardiomyopathy. the images are impressive and the work done is worth compliments. My main concern is how you have grouped the animals. While the diagnosis of hyperthyroidism is straightforward in the alive cat thanks to specific biochemical analysis, the diagnosis of primary hypertrophic cardiomyopathy is more complicated. Therefore, in your group of cats with hyperthyroidism, you cannot exclude a concurrent primary HCM based on clinical criteria in the alive animals. Considering you have the histopathology, I would strongly advise that you make use of these results to contribute to proper grouping. The Materials and Methods need to reflect this and describe inclusion/exclusion criteria in detail. See below a more detailed review with line numbers specified.
Line 23, 34, 56: Please refer to hypertrophic cardiomyopathy phenotype, or to concentric hypertrophy.
Line 24: consider using “but structural myocardial alterations persist”
Line 55-58: these chances are not always present in FHT, and if present, they do not always lead to congestive heart failure. Please rephrase to reflect the reality better.
Line 63-64: “Although in cats with treated FHT the left ventricular wall and interventricular septum present normal thickness” this is also not always true, with different responses reported, also depending on the type of treatment. Also, please provide a reference. If this is also 10, then refer to this particular study more clearly.
Line 85-86: “Research also suggests the role of calreticulin overexpression in cardiac fibrosis”. If you say the role, you should explain which role. Otherwise, maybe better state “a role”.
Line 90-91: “promotes the stiffness of the heart” it is not clear to mean what this means.
Line 108-112: Because grouping the animals is central in your study, it needs to be described better which were the clinical criteria to divide them in such groups. Please describe thoroughly how hyperthyroidism was diagnosed, how primary HCM was diagnosed, and which inclusion criteria did the control have to meet. More specifically, for the diagnosis of HCM, you mainly refer to the ACVIM consensus statement. Should we assume that all these cats had also the the T4 and blood pressure measured and maybe even IGF-1? Was the diagnosis based on echocardiography? But echocardiography cannot not provide a definitive diagnosis. Considering you have histopathology, should you not use this technique also for grouping the correctly? Based on the presence of myocardial disarray? Are there criteria available for definitive diagnosis? You refer to your previous study, there also there was a significant difference in the degree of disarray. Should that not be a criterion for classification of the cats in the different groups?
Line 258-260: is this something reported in humans? Can you specify? And in heart failure means in patients congestive heart failure? Or what do you exactly mean?
Line 264-266: “Instead, it is more related to cardiac hypertrophy, cardiomegaly, left atrial enlargement and compromised blood supply to cardiomyocytes due to arterial narrowing than to primary disease”. I this this sentence is too assertive. It is still an hypothesis, your study did not aim to answer this question and we don’t know if the number of specimens is enough to conclude this. I would keep it as “the results suggest that it might be related to…”
Line 275-276: are we sure that cats with FHTH did not have a concurrent primary HCM? You mention this limitation at lines 314-316, but it needs to be expanded.
Line 278-284: You state that calreticulin has a protective effect to then say it induces cardiomyopathy based on another reference. It seems controversial. Can you review/clarify better?
Line 284-289: It is not clear what you want to say about the calreticulin and the interstitial tissue here. I understand that in your study calreticulin was only present in the interstitial tissue in cats with HCM. Then you report how the interstitial tissue increases in heart failure, were the cats of your study in heart failure? Did you also encounter more interstitial tissue than expected?
Line 294-295: “Calreticulin immunoreaction in the current research shown none significant correlation with myocardial fibrosis”. I miss the description of this statistical evaluation in the Materials and Methods.
Line 301-302: “In the current study no relation between the 301 IL-10 expression and the extent of fibrosis was noted”. Similar to above, it is not lcear from Materials and Methods what you are actually testing. And in the Results section, there is no mention about correlation with fibrosis.
Line 341-316: this is a big limitation of the study and needs to be given more importance/be expanded. Furthermore, the ACVIM guidelines are reported incorrectly. In case of an HCM phenotype on echocardiography, one must exclude hyperthyroidism to diagnose a primary HCM. But if hyperthyroidism is diagnosed, it is not stated that primary HCM cannot be concurrent. Please amend.
“These changes appear to be more closely associated with secondary processes such as cardiac hypertrophy, cardiomyocyte degeneration and reduced myocardial perfusion, rather than with the primary aetiology of myocardial disease”. These conclusions are influenced by the limitations of your study, so they can only be suggested, this sentence seems to assertive.
Author Response
Dear Reviewer,
We would like to thank you for all the comments sent.
In response, we have made appropriate changes in the text, and responded individually to each point you made. All changes are marked red in the new version of the manuscript.
We also attach a detailed answer to all the comments.
Dear authors,
Your manuscript contains important and unique information about cardiac pathology in feline hyperthyroidism/hypertrophic cardiomyopathy. the images are impressive and the work done is worth compliments.
Thank you for your appreciation of our work.
My main concern is how you have grouped the animals. While the diagnosis of hyperthyroidism is straightforward in the alive cat thanks to specific biochemical analysis, the diagnosis of primary hypertrophic cardiomyopathy is more complicated. Therefore, in your group of cats with hyperthyroidism, you cannot exclude a concurrent primary HCM based on clinical criteria in the alive animals. Considering you have the histopathology, I would strongly advise that you make use of these results to contribute to proper grouping. The Materials and Methods need to reflect this and describe inclusion/exclusion criteria in detail. See below a more detailed review with line numbers specified.
Thank you for that comment. We agree that proper grouping of animals is essential for our study. Nonetheless, although there are some histopathological changes that suggest HCM, the histopathological picture of the disease is very complex and the changes often overlap with other diseases (e.g. hyperthyroidism, systemic hypertension) what was also presented in other papers. Also disarray, although strongly suggestive of HCM, can also be present in cardiac hypertrophy of other cause (e.g. https://doi.org/10.1016/j.jcpa.2024.11.006). In our opinion, HCM can be diagnosed only with a complete clinical history and with cardiac hypertrophy confirmed on the necropsy with the exclusion of other causes of hypertrophy or pseudohypertrophy, like myocarditis or neoplastic disease. Therefore, in our study we designed the groups basing on the clinical symptoms and additional examination (blood tests, pressure measurements, echocardiography). We chose the cases carefully to meet the inclusion criteria, what resulted in a limited number of cases in each group. We added the information in the Materials and methods section and in the limitations of our study. We also expanded the inclusion/exclusion criteria description to accurately describe the study design.
Line 23, 34, 56: Please refer to hypertrophic cardiomyopathy phenotype, or to concentric hypertrophy.
Phrases changed as suggested.
Line 24: consider using “but structural myocardial alterations persist”
Phrases changed as suggested.
Line 55-58: these chances are not always present in FHT, and if present, they do not always lead to congestive heart failure. Please rephrase to reflect the reality better.
We rephrased the paragraph to be more consistent with the reality.
Line 63-64: “Although in cats with treated FHT the left ventricular wall and interventricular septum present normal thickness” this is also not always true, with different responses reported, also depending on the type of treatment. Also, please provide a reference. If this is also 10, then refer to this particular study more clearly.
Thank you for that comment. This sentence was not a general comment but was based on our previous study and referred only to that study. We rephrased the sentence and added reference to make it clearer.
Line 85-86: “Research also suggests the role of calreticulin overexpression in cardiac fibrosis”. If you say the role, you should explain which role. Otherwise, maybe better state “a role”.
Thank you for that comment. We changed the sentence as suggested.
Line 90-91: “promotes the stiffness of the heart” it is not clear to mean what this means.
We rephrased the sentence for clarification.
Line 108-112: Because grouping the animals is central in your study, it needs to be described better which were the clinical criteria to divide them in such groups. Please describe thoroughly how hyperthyroidism was diagnosed, how primary HCM was diagnosed, and which inclusion criteria did the control have to meet. More specifically, for the diagnosis of HCM, you mainly refer to the ACVIM consensus statement. Should we assume that all these cats had also the the T4 and blood pressure measured and maybe even IGF-1? Was the diagnosis based on echocardiography? But echocardiography cannot not provide a definitive diagnosis. Considering you have histopathology, should you not use this technique also for grouping the correctly? Based on the presence of myocardial disarray? Are there criteria available for definitive diagnosis? You refer to your previous study, there also there was a significant difference in the degree of disarray. Should that not be a criterion for classification of the cats in the different groups?
Again, thank you for that comment. We changed the description of inclusion/exclusion criteria and added more detail. As mentioned in our previous response, there are no clear histopathological criteria to diagnose HCM. The pathological picture of HCM is very complex (https://doi.org/10.3390/ani15050703), and even the most typical signs like cardiomyocyte hypertrophy or disarray can be present in hypertrophy of other origin (like hypertension: https://doi.org/10.1016/j.jcpa.2024.11.006 or hyperthyroidism: https://doi.org/10.1080/01652176.2023.2234436). And although the difference between in disarray presence was significant between HCM and FHT cats, also cats with FHT may present with disarray, therefore, in our opinion it cannot be a clear cut-off criterium.
Line 258-260: is this something reported in humans? Can you specify? And in heart failure means in patients congestive heart failure? Or what do you exactly mean?
Thank you for pointing out this ambiguity. The findings were reported in dogs and humans in DCM and in dogs with experimentally-induced ischemia-related progressive heart failure. We added the information to the manuscript.
Line 264-266: “Instead, it is more related to cardiac hypertrophy, cardiomegaly, left atrial enlargement and compromised blood supply to cardiomyocytes due to arterial narrowing than to primary disease”. I this this sentence is too assertive. It is still an hypothesis, your study did not aim to answer this question and we don’t know if the number of specimens is enough to conclude this. I would keep it as “the results suggest that it might be related to…”
Thank you for that comment. We changed the sentence, as suggested.
Line 275-276: are we sure that cats with FHTH did not have a concurrent primary HCM? You mention this limitation at lines 314-316, but it needs to be expanded.
As some of the cats present with cardiac concentric hypertrophy despite FHT treatment, we cannot rule out a concomitant FHT and HCM. Also, as mentioned before, at present, there are no clear cut-off histological criteria for HCM and FHT. Therefore, we propose to divide the cats into HCM and FHT group basing on clinical information. It would be beneficial to compare the results in cats with FHT with and without cardiac concentric hypertrophy but the number of samples was too low to perform reliable statistical analysis. We put that information in the limitations of our study.
Line 278-284: You state that calreticulin has a protective effect to then say it induces cardiomyopathy based on another reference. It seems controversial. Can you review/clarify better?
Thank you for that comment. Indeed, the role that calreticulin can play in adult heart is complex. On the one hand it shows a protective effect on cardiomyocytes; on the other hand, acting on endoplasmic reticulum function, it enhances cardiomyocyte strength, that can lead to stretching of fibroblasts and eventually to fibrosis. We added explanation in the text.
Line 284-289: It is not clear what you want to say about the calreticulin and the interstitial tissue here. I understand that in your study calreticulin was only present in the interstitial tissue in cats with HCM. Then you report how the interstitial tissue increases in heart failure, were the cats of your study in heart failure? Did you also encounter more interstitial tissue than expected?
Thank you for that comment. All cats with HCM were in C stage disease with acute or severe signs of heart failure, while cats with FHT were more diverse in that manner: some presented with heart failure while some had good cardiac function. That difference may explain why only cats with HCM showed significantly higher amount of calreticulin when compared to control group. Nonetheless, no relation of the amount of calreticulin to cardiac fibrosis was noted in the studied animals. We expanded the paragraph to make it clearer.
Line 294-295: “Calreticulin immunoreaction in the current research shown none significant correlation with myocardial fibrosis”. I miss the description of this statistical evaluation in the Materials and Methods.
The information that the correlation between the immunohistochemical reaction and pathomorphological analysis was performed was already in the Materials and methods section. To clarify the extent of statistical analysis, we added the information that the pathomorphological examination involves both gross and histopathological examination. We hope that this description is sufficient and that it is not needed to list all the parameters that were corelated with each other. But if the Reviewers think it would be beneficial to the readers and clarify the methodology of the study, we can put that information in the next version of the manuscript.
Line 301-302: “In the current study no relation between the 301 IL-10 expression and the extent of fibrosis was noted”. Similar to above, it is not lcear from Materials and Methods what you are actually testing. And in the Results section, there is no mention about correlation with fibrosis.
As mentioned above, we added information to the Materials and method section. In the previous version of the manuscript, we did not report the correlations that showed no significance. But to avoid confusion, we added information to the results section that fibrosis showed no correlation with antibodies reaction.
Line 341-316: this is a big limitation of the study and needs to be given more importance/be expanded. Furthermore, the ACVIM guidelines are reported incorrectly. In case of an HCM phenotype on echocardiography, one must exclude hyperthyroidism to diagnose a primary HCM. But if hyperthyroidism is diagnosed, it is not stated that primary HCM cannot be concurrent. Please amend.
Thank you for that comment. We expanded the limitation section, as suggested.
“These changes appear to be more closely associated with secondary processes such as cardiac hypertrophy, cardiomyocyte degeneration and reduced myocardial perfusion, rather than with the primary aetiology of myocardial disease”. These conclusions are influenced by the limitations of your study, so they can only be suggested, this sentence seems to assertive.
Thank you for that comment. We changed the conclusions to be in accordance with the results and limitations of our study.
Sincerely,
Authors
Reviewer 2 Report
Comments and Suggestions for Authors
I thank the authors for conducting this insightful study on the expression of immunohistochemical markers in control cats, as well as in cats with hyperthyroidism and hypertrophic cardiomyopathy. Exploring differences in expression levels and potential correlations enhances our understanding of these two prevalent conditions in feline clinical practice.
Below are several concerns and suggestions that, if addressed, may strengthen the manuscript:
Abbreviations: From the moment an abbreviation is introduced, it must be used in the text (e.g. hypertrophic cardiomyopathy).
Although the authors refer to a previous study (number 10), we recommend including a brief description of the clinical criteria used to establish the three groups of cats in the current manuscript. This will provide essential context and ensure the study is interpretable as a standalone work.
Clarification is needed regarding the diagnostic procedures used to confirm the health status of the control group. Specifically:
- Was echocardiography performed to exclude subclinical cardiomyopathy or other cardiac abnormalities?
- Were systemic hypertension and hyperthyroidism actively ruled out? Given that both conditions may be asymptomatic and influence cardiac structure and function, detailing how these were assessed is critical to validate the control group.
It would be helpful to indicate the clinical stage of cats in the HCM group, namely, whether they were asymptomatic (stages B1/B2) or symptomatic (stage C), to better understand how the presence or absence of heart failure may have influenced the immunohistochemical findings.
We recommend including the morphometric cardiac data for each group within the present manuscript. This is particularly important because the number of cats differs from the authors’ previous study, which is referenced as the source of morphometric data for statistical analysis in this work.
Please clarify whether the cats with hyperthyroidism exhibited concentric left ventricular hypertrophy. This detail is essential, as the conclusions regarding the shared alteration in desmin and IL-10 expression profiles between the FHT and HCM groups also rely on the presence of hypertrophic changes in both conditions.
While the median calreticulin-positive area appears elevated in HCM group, the range and boxplot suggest considerable overlap with control group. A similar pattern is seen for IL-10 (including FHT group). Please review and clarify the interpretation of these findings in Figure 5 to ensure alignment between the data and the conclusions drawn.
What are the potential pathophysiological explanations for the observation that increased calreticulin expression was found only in the HCM group and not in the FHT group? A discussion of this finding would add important depth to the interpretation of the results.
In the Discussion section, the authors refer no correlation between the expression of markers (calreticulin, IL-10) and the presence of myocardial fibrosis; however, the corresponding data are not presented in the manuscript. Please review and include these results to support the statements made.
Author Response
Dear Reviewer,
We would like to thank you for all the comments sent.
In response, we have made appropriate changes in the text, and responded individually to each point you made. All changes are marked red in the new version of the manuscript.
We also attach a detailed answer to all the comments.
I thank the authors for conducting this insightful study on the expression of immunohistochemical markers in control cats, as well as in cats with hyperthyroidism and hypertrophic cardiomyopathy. Exploring differences in expression levels and potential correlations enhances our understanding of these two prevalent conditions in feline clinical practice.
Thank you for your appreciation of our work.
Below are several concerns and suggestions that, if addressed, may strengthen the manuscript:
Abbreviations: From the moment an abbreviation is introduced, it must be used in the text (e.g. hypertrophic cardiomyopathy).
We revised the text and used the abbreviations, where appropriate. Only names appearing at the beginning of sentence were left in the full version.
Although the authors refer to a previous study (number 10), we recommend including a brief description of the clinical criteria used to establish the three groups of cats in the current manuscript. This will provide essential context and ensure the study is interpretable as a standalone work.
Thank you for that comment. We added an inclusion/exclusion criteria description in the Materials and methods section.
Clarification is needed regarding the diagnostic procedures used to confirm the health status of the control group. Specifically:
- Was echocardiography performed to exclude subclinical cardiomyopathy or other cardiac abnormalities?
- Were systemic hypertension and hyperthyroidism actively ruled out? Given that both conditions may be asymptomatic and influence cardiac structure and function, detailing how these were assessed is critical to validate the control group.
Thank you for that comment. We added appropriate information in the inclusion/exclusion criteria for each group.
It would be helpful to indicate the clinical stage of cats in the HCM group, namely, whether they were asymptomatic (stages B1/B2) or symptomatic (stage C), to better understand how the presence or absence of heart failure may have influenced the immunohistochemical findings.
All cats in HCM group were symptomatic at time of death or euthanasia. We added appropriate information to the Materials and methods section.
We recommend including the morphometric cardiac data for each group within the present manuscript. This is particularly important because the number of cats differs from the authors’ previous study, which is referenced as the source of morphometric data for statistical analysis in this work.
Thank you for that comment. We added the morphometric data to the results section, as suggested. We did not add the histopathological data but if the Reviewers consider them essential, we will add them in the next version of the manuscript.
Please clarify whether the cats with hyperthyroidism exhibited concentric left ventricular hypertrophy. This detail is essential, as the conclusions regarding the shared alteration in desmin and IL-10 expression profiles between the FHT and HCM groups also rely on the presence of hypertrophic changes in both conditions.
Some of the cats (37.5%) in FHT group presented with concentric hypertrophy. A small number of animals in that group did not allow to make a reliable statistical analysis with FTH subgroups: with or without hypertrophy. We added appropriate information to the Results and Discussion sections.
While the median calreticulin-positive area appears elevated in HCM group, the range and boxplot suggest considerable overlap with control group. A similar pattern is seen for IL-10 (including FHT group). Please review and clarify the interpretation of these findings in Figure 5 to ensure alignment between the data and the conclusions drawn.
Thank you for that comment. We added information to the description of the results.
What are the potential pathophysiological explanations for the observation that increased calreticulin expression was found only in the HCM group and not in the FHT group? A discussion of this finding would add important depth to the interpretation of the results.
Thank you for that comment. Calreticulin showed enhanced expression in both FHT and HCM groups, although only in HCM group it reached significance. It may be caused by a more diversity in FHT group with both normal and hypertrophied hearts. We added the information to the discussion section.
In the Discussion section, the authors refer no correlation between the expression of markers (calreticulin, IL-10) and the presence of myocardial fibrosis; however, the corresponding data are not presented in the manuscript. Please review and include these results to support the statements made.
In the previous version of the manuscript we only included correlations that were statistically significant. To avoid confusion, we added information about the lack of significant correlations between fibrosis and antibodies reactions.
Sincerely,
Authors
Round 2
Reviewer 1 Report
Comments and Suggestions for Authors
Thank you for amending the manuscript accurately, and for the thorough answers to my comments.
Author Response
Dear Reviewer,
Thank you for appreciating our work in improving the manuscript.
Sincerely,
Authors
Reviewer 2 Report
Comments and Suggestions for Authors
The authors have thoroughly and effectively addressed the reviewers' comments.
Just 3 minor comments:
-Line 37: the abbreviation HCM should be introduced in line 34.
-Line 44: Given that a percentage (37.5 per cent) of cats with hyperthyroidism had ventricular hypertrophy, what do the authors mean by “despite normal left ventricular dimensions”?
-Line 67: the abbreviation HCM should be introduced in line 57.
Author Response
Dear Reviewer,
Thank you for appreciating our work in improving the manuscript.
Below we attach responses for current comments. All changes are also highlighted red in the text.
-Line 37: the abbreviation HCM should be introduced in line 34.
We made changes as suggested
-Line 44: Given that a percentage (37.5 per cent) of cats with hyperthyroidism had ventricular hypertrophy, what do the authors mean by “despite normal left ventricular dimensions”?
Thank you for that comment. 37.5% of cats showed cardiac hypertrophy (cardiac mass >20g) but median value for left ventricular wall thickness and interventricular septum thickness were not significantly different from the normal group. Also, the cardiac hypertrophy was not severe in that group. We changed the sentence to make it more clear and added "mild" to "left ventricular hypertrophy" in line 203.
-Line 67: the abbreviation HCM should be introduced in line 57.
We made changes as suggested
Sincerely,
Authors